# Twisted oxide lateral homostructures with conjunction tunability

Ping-Chun Wu[1,7], Chia-Chun Wei[1,7], Qilan Zhong[2,7], Sheng-Zhu Ho[1], Yi-De Liou[1], Yu-Chen Liu[1], Chun-Chien Chiu[1], Wen-Yen Tzeng[3], Kuo-En Chang[1], Yao-Wen Chang[1], Junding Zheng[2], Chun-Fu Chang [4], Chien-Ming Tu[3], Tse-Ming Chen [1], Chih-Wei Luo [3,5], Rong Huang [2], Chun-Gang Duan[2], Yi-Chun Chen [1], Chang-Yang Kuo[3,5] & Jan-Chi Yang [1,6✉]

Epitaxial growth is of significant importance over the past decades, given it has been the key process of modern technology for delivering high-quality thin films. For conventional het-eroepitaxy, the selection of proper single crystal substrates not only facilitates the integration of different materials but also fulfills interface and strain engineering upon a wide spectrum of functionalities. Nevertheless, the lattice structure, regularity and crystalline orientation are determined once a specific substrate is chosen. Here, we reveal the growth of twisted oxide lateral homostructure with controllable in-plane conjunctions. The twisted lateral homo-structures with atomically sharp interfaces can be composed of epitaxial "blocks" with dif-ferent crystalline orientations, ferroic orders and phases. We further demonstrate that this approach is universal for fabricating various complex systems, in which the unconventional physical properties can be artificially manipulated. Our results establish an efficient pathway towards twisted lateral homostructures, adding additional degrees of freedom to design epitaxial films.

[1] Department of Physics, National Cheng Kung University, Tainan 70101, Taiwan. [2] Key Laboratory of Polar Materials and Devices (MOE) and Department of Electronics, East China Normal University, 200241 Shanghai, China. [3] Department of Electrophysics, National Yang Ming Chiao Tung University, Hsinchu 30010, Taiwan. [4] Max-Planck Institute for Chemical Physics of Solids, Dresden 01187, Germany. [5] National Synchrotron Radiation Research Center, Hsinchu 30076, Taiwan. [6] Center for Quantum Frontiers of Research & Technology (QFort), National Cheng Kung University, Tainan 70101, Taiwan. [7] These authors contributed equally: Ping-Chun Wu, Chia-Chun Wei, Qilan Zhong. ✉email: janchiyang@phys.ncku.edu.tw

Epitaxial growth, by which desired materials are deposited on crystalline substrates with well-aligned features, has been widely utilized to manufacture high-quality thin films with excellent functionalities, playing an indispensable role no matter from technological or scientific point of view. In conventional heteroepitaxy, the first consideration is the lattice mismatch between the desired materials and the single crystal substrates, in which the caused epitaxial strain in the grown thin films would lead to direct modification of physical properties[1], crystalline phases, and the formation of defects[2]. Over the past decades, heteroepitaxy has been adopted to fabricate complex heterostructures, such as superlattices and vertically aligned arrays[3,4]. In these epitaxial systems, taking advantages of the sophisticated interactions between different materials, unconventional physical properties which cannot be obtained from individual parent materials are usually found[5,6]. Hitherto, heteroepitaxy is still a very broad and rapidly developing field, in which researchers are dedicated to comprehensive understanding and advanced modulation of exotic phenomena in epitaxial thin films, nanostructures, and superlattices[7,8]. Nevertheless, in contrast to conventional heteroepitaxy which mainly focuses on vertical stacking, the development of epitaxial lateral homostructures has faced significant bottlenecks.

The precise control of epitaxial lateral homostructures provides additional degrees of freedom to create twisted interfaces/junctions and manipulate topological defects such as phase boundaries[9], domain walls[10] and vortices[11], offering unique platforms for revealing emergent physical properties and developing technically important nanoelectronics. To fabricate lateral homostructures, strategies based on thin film deposition on artificially modified substrates, such as bi(tri)crystalline substrates[12,13], step-edge template[14–16], and bi-epitaxy growth[16–19] were proposed. Additionally, hybrid-orientation technology (HOT)[20] that utilizes modern semiconductor fabrication procedures including hydrogen implantation, a combination of starting wafer and handle wafer, split anneal, and chemical mechanical polishing has also established a landmark for constructing lateral junctions composed of different crystal orientations. From a traditional epitaxial growth perspective, the epitaxial thin films are grown following the lattice patterns provided by the chosen single crystal substrates and are constrained to exhibit similar crystal geometries. In this regard, the major advantage of heteroepitaxy using pristine single crystal substrates becomes a shortcoming when it comes to the fabrication of lateral homostructures. Despite the fact that the aforementioned methods employing artificially modified substrates are indeed practical, fabricating twisted lateral homostructures with geometry scalability and repetitively alternating configurations remains extremely challenging, especially for strongly correlated systems.

In this work, using multiferroic BiFeO$_3$ (BFO) as a model system, we reveal a generic method assisted by inserting a freestanding oxide layer with controllable twist angle and lattice pattern to fulfill the efficient growth and manipulation of the twisted epitaxial lateral homostructures. In addition, we manufactured a twisted oxide lateral homostructure based on a colossal magnetoresistance manganite, La$_{0.7}$Sr$_{0.3}$MnO$_3$ (LSMO), to demonstrate the capability to construct repeated alternating orbital arrays. The results suggest that our approach is generic for fabricating a wide spectrum of complex systems, via which unusual physical properties such as enhanced second harmonic generation and exotic magnetotransport behaviors can be derived. The proposed approach paves a platform for the development of lateral homostructures with desired twist angle, crystal orientation, and phase conjunction degrees of freedom, thus opening up a distinct scene for epitaxial growth.

## Results and discussion

The process flow of proposed twisted lateral homostructures is illustrated in Fig. 1a. To begin with, an ultrathin SrTiO$_3$ (STO) thin film and a LSMO sacrificial layer were grown on (110)-oriented STO single crystal substrate via pulsed laser deposition (see "Methods"). The STO/LSMO/STO heterostructure was then immersed in an acid solution, which is used to dissolve the LSMO layer, separating the grown STO thin film and STO substrate. A similar freestanding process via utilizing LSMO as a sacrificial layer has been reported by Bakaul et al.[21] and Shen et al.[22] The freestanding (110)-oriented STO layer (FS-STO) was then obtained and transferred onto the other (110)-oriented STO substrate. At the boundary of the covered and uncovered region, the [1$\bar{1}$0] directions of FS-STO and STO substrate form a natural but adjustable twist angle (denoted as $\varphi$), as shown in Fig. 1a. A (110)-oriented STO twisted template with misaligned in-plane crystalline directions is therefore derived. Herein, a classic room-temperature multiferroic material with coupled ferroelectricity and antiferromagnetism, BFO, was chosen as a model system for exploring the conjunction tunability. When it comes to epitaxial growth of BFO, the topography and ferroelectric domain patterns are determined by constrains and orientation applied by the substrate beneath. The special stripe feature in epitaxial (110)-oriented BFO is therefore a direct visual evidence once the crystalline directions are altered. Figure 1b shows the topography images of BFO grown on pristine STO region (BFO$_{AG}$), near the boundary and on FS-STO (BFO$_{FS}$), respectively. Given that the topography stripes in (110)-oriented BFO are always aligned with the [001] direction of STO[23], the twisted feature of the grown BFO at the boundary can be clearly observed. Please be noted that the twisted feature of the epitaxial (110)-oriented BFO can also be resolved by X-ray phi scan, as shown in Supplementary Note 1 and Supplementary Fig. 1. Noticeably, the BFO grown on top of the twisted template follows the underlying crystalline patterns of the FS-STO layer and the STO substrate, respectively, forming a twisted lateral homostructure/homojunction. This suggests that the grown epitaxial BFO films tend to follow crystalline architectures of the freestanding layer and single-crystal substrate right beneath, which is in good agreement with our density-functional theory (DFT) calculations (Supplementary Note 2 and Supplementary Fig. 2).

To reveal the detailed structure of the lateral homostructure grown on the twisted template, the microstructure of a well-defined 90º twisted (110)-oriented BFO homostructure was studied via transmission electron microscopy (TEM). A typical low magnification cross-sectional bright-field TEM image along the [001]$_{STO}$ direction (Fig. 2a) shows the clear boundaries at the STO/BFO$_{FS}$/BFO$_{AG}$ interface. The corresponding selected area electron diffraction (SAED) patterns of the STO substrate, BFO$_{AG}$ and BFO$_{FS}$ are shown in the insets, respectively. The epitaxial relationships of STO(110) // BFO$_{AG}$(110) // BFO$_{FS}$(110) and STO[001] // BFO$_{AG}$[001]// BFO$_{FS}$[1$\bar{1}$0] are revealed. The typical high-resolution transmission electron microscopy (HRTEM) images viewed along the STO [001] zone axis (Fig. 2b) and STO [1$\bar{1}$0] zone axis (Fig. 2c) show the detailed structures of the interface region. It can be clearly seen that the BFO$_{AG}$ is epitaxially grown on the STO substrate, while the BFO$_{FS}$ is epitaxially grown on a brighter ~5 nm STO$_{FS}$ layer as marked between two green lines. In addition, there is a tiny gap between the FS-STO and STO substrate, which is filled with BFO during the PLD growth. The twisted BFO$_{AG}$/BFO$_{FS}$ interface is very sharp as marked by the yellow dashed lines. These SAED and HRTEM results unanimously manifest the high quality of the 90º twisted BFO homostructure. It is remarkable that the twist angle ($\varphi$) of the homostructures can be arbitrarily controlled through the alignment process during transfer. From the surface morphology near the boundary, the relationships of the [001]-oriented stripes of (110)-oriented BFO homostructures can be

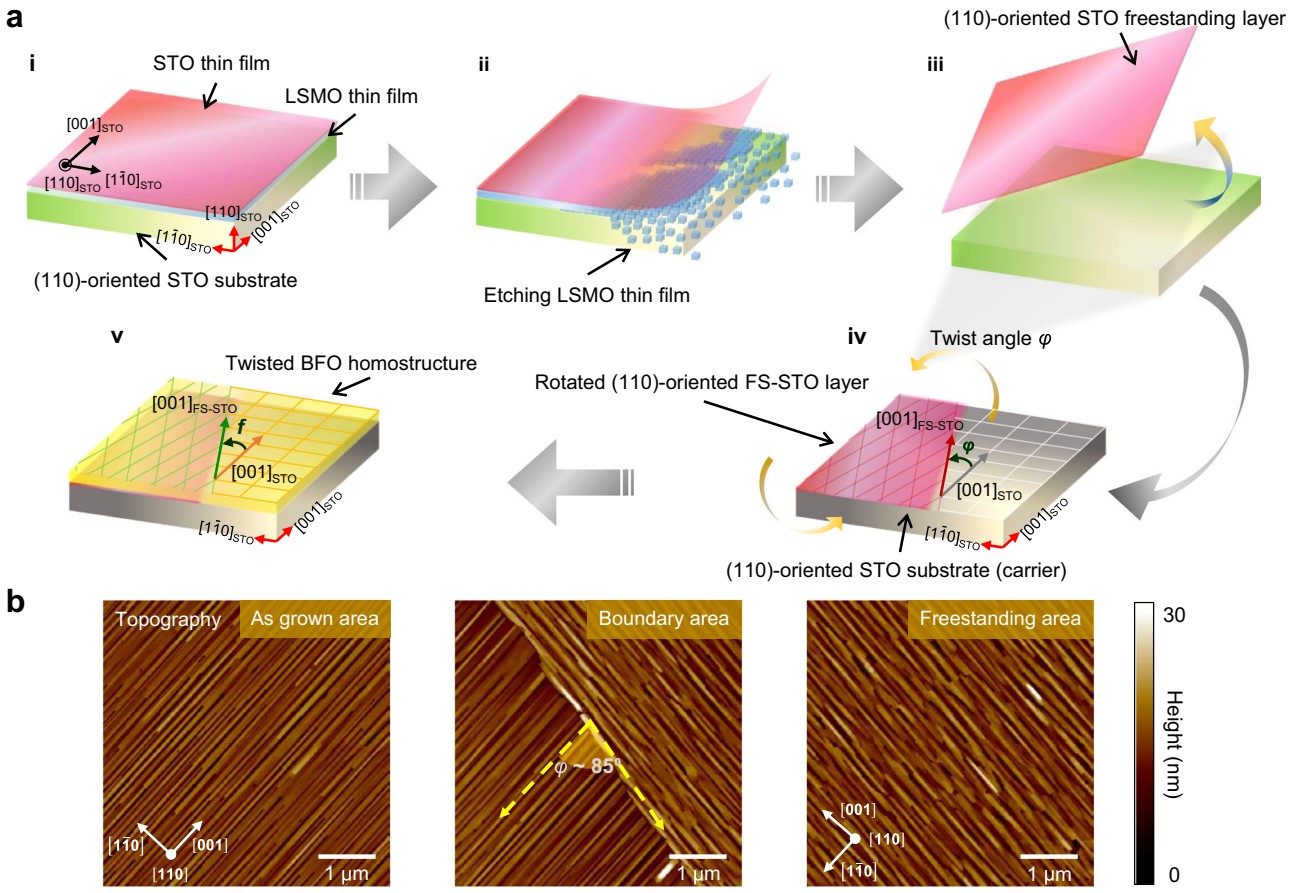

**Fig. 1 Synthesis of the lateral oxide homostructures. a** Process flow for fabricating twisted lateral oxide homostructures. (i) Growth of STO with a LSMO sacrificial layer on (110)-oriented STO single crystal substrate. (ii) Etching of the LSMO sacrificial layer. (iii) Releasing and (iv) transferring the freestanding STO layer with designated twist angle with respect to the selected single-crystal carrier substrates. (v) Deposition of BFO thin film on the twisted template. **b** Topography images of (110)-oriented BFO grown on pristine STO, boundary, and FS-STO region, respectively. The $[001]_{pc}$-oriented stripes serve as the direction reference of epitaxial BFO, identifying the in-plane directions of (110)-oriented BFO and the twist angle (denoted as angle $\varphi$) at the boundary of pristine STO and FS-STO region.

easily identified (see Supplementary Note 3 and Supplementary Fig. 3).

Furthermore, to understand the atomic arrangement at the boundary of $BFO_{AG}$ region and $BFO_{FS}$ region, the high-angle annular dark-field (HAADF) observations were carried out in a spherical aberration-corrected scanning TEM (Cs-corrected STEM), as shown in Fig. 2d, e, respectively. When viewed along the STO [001] direction, the $BFO_{AG}$ also aligns to the [001] zone axis, showing a typical perovskite structure. The $BFO_{AG}$ (010) plane is about 45° with respect to the STO substrate surface and clear facets of the $BFO_{AG}$ (010) plane at the $BFO_{AG}/BFO_{FS}$ interface can be seen, as indicated with a yellow dashed line. However, when observed along the STO [1$\bar{1}$0] zone axis (Fig. 2e), which is perpendicular to the STO [001] zone axis, the $BFO_{AG}/BFO_{FS}$ interface became nearly perpendicular to the STO substrate surface. This indicates that the facets of FS-STO and $BFO_{AG}$ (001) planes governed the conjunction geometry of the $BFO_{AG}/BFO_{FS}$ interface, as marked with the yellow dashed lines in Figs. 2d, e. In order to reveal the detailed interfacial microstructure of $BFO_{AG}$ and $BFO_{FS}$, the interface regions in Fig. 2e were enlarged and overlaid with the structural model of BFO, as shown in Fig. 2f, g. A transition monolayer can be seen at the flat interface region, as indicated by the orange dashed box in Fig. 2f. The contrast of this monolayer is weaker than pure Bi columns and is stronger than that of the pure Fe columns in BFO, exhibiting as a mixed transition layer. In contrast, a clear atomic step

is observed in Fig. 2g, as marked by the red circle, where a strong structure distortion can be observed. Through the combination of the alternating flat and step interfaces at the atomic scale, the large misalignment between $BFO_{AG}$ and $BFO_{FS}$ can be accommodated. Figure 2h, i shows the electron energy-loss near-edge structure (ELNES) of O-K and $Fe-L_{2,3}$ from point 1 ($BFO_{FS}$), point 2 (interface) and point 3 ($BFO_{AG}$), respectively. The O-K edge from the interface (point 2) clearly shows that the peak a is lower than that of BFO (points 1 and 3), indicating the existence of oxygen vacancy in the interface region[24]. Correspondingly, the $Fe-L_3$ edge from the interface region shows a chemical shift to the lower energy side, as indicated by an arrow in Fig. 2i, indicating that the valence state of Fe decreased at the interface[25]. Additionally, atomic resolved X-ray energy dispersive spectroscopy (EDS) mapping of the interface region was performed (Supplementary Fig. 4), which further confirmed the high-quality BFO homojunction.

Utilizing the ability to combine different crystallographic orientations in the in-plane, a device prototype composed of epitaxial BFO blocks with designated ferroelectric polarizations and antiferromagnetic directions is demonstrated, as schematically illustrated in Fig. 3a. Firstly, FS-STO (110) was transferred onto STO (110) substrate, forming an in-plane twist angle of 90° with respect to their individual [001] directions. Through a combination of photolithography and etching processes (see Methods), the FS-STO was then etched into well-defined patterns. Last, as

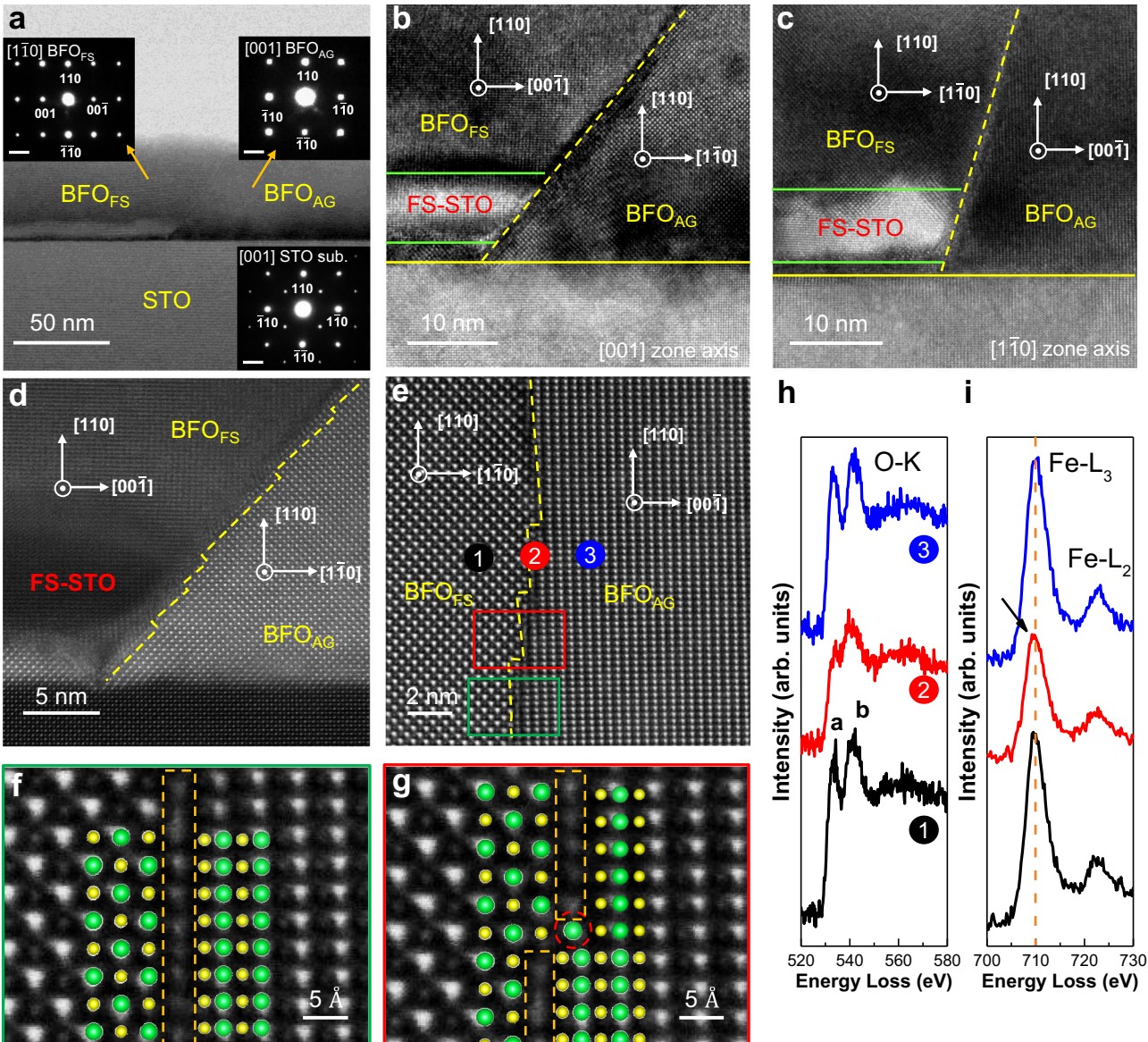

**Fig. 2 Microstructure characterization of the 90° twisted BFO lateral homostructure. a** Typical cross-sectional TEM image taken along [001] direction of STO substrate. The insets show the corresponding SAED patterns of STO substrate, BFO grown on pristine STO (BFO$_{AG}$), and BFO grown on FS-STO (BFO$_{FS}$), respectively, in which the scale bar refers to 2 nm$^{-1}$. **b, c** HRTEM images of the interface area taken along the [001] and [1$\bar{1}$0] direction of STO substrate, respectively. **d** HAADF image of the interface area observed along the [001] direction of STO substrate. A gap between FS-STO and STO can be clearly seen, where the space was further filled with BFO. **e** Typical HAADF image of the BFO$_{FS}$/BFO$_{AG}$ interface. **f, g** Enlarged atomic images from the green and red rectangles in (**e**). **h** O-K and (**i**) Fe-L$_{2,3}$ edges acquired from the region 1, 2, and 3 in (**e**).

above-mentioned, BFO thin film was then grown on the patterned twisted template. Following similar flows, any patterned lateral homostructures with relative in-plane twist angle can be fabricated. As shown in Fig. 3b, the topography image reveals the vertical stripes embedded in the horizontal stripe matrix, verifying the feasibility of arbitrary control of lateral conjunctions. Besides, vector piezoresponse force microscopy (vector-PFM) analysis with respect to different cantilever orientation was carried out to identify the ferroelectric domain patterns of the cellular structure. As shown in Fig. 3c, the in-plane PFM phase image was taken with cantilever pointing along the [1$\bar{1}$0] direction of pristine STO substrate. For patterned squares inside the matrix, a single domain feature with net ferroelectric polarization perpendicular to the cantilever is observed, while the net ferroelectric polarization of the matrix is parallel to the cantilever. The perpendicular

polarization configuration could also be identified by rotating the probing cantilever 45° counterclockwise, as shown in Fig. 3d.

In multiferroic BFO, the relationship between the spontaneous polarization and antiferromagnetism is subject to epitaxial constrains[26,27]. With the development of twisted lateral homostructures, the arrangement of ferroic orders can be controlled in a more flexible and innovative manner. To investigate the antiferromagnetic characteristics of the patterned BFO (110) lateral device, we performed X-ray linear absorption spectroscopy (XLAS) measurements on the Fe L$_{2,3}$-edge of the BFO$_{AG}$ and BFO$_{FS}$ thin film with a linear polarization vector (**E**) paralleled to the pristine STO [001]$_{SUB}$. Fig. 3e shows the XLAS mapping of intensity ratio I$_B$/I$_A$, where I$_A$ and I$_B$ are the XLAS peak intensity taken at Fe L$_2$-edge around the characteristic energy denoted as A and B, respectively, as indicated in Fig. 3f.

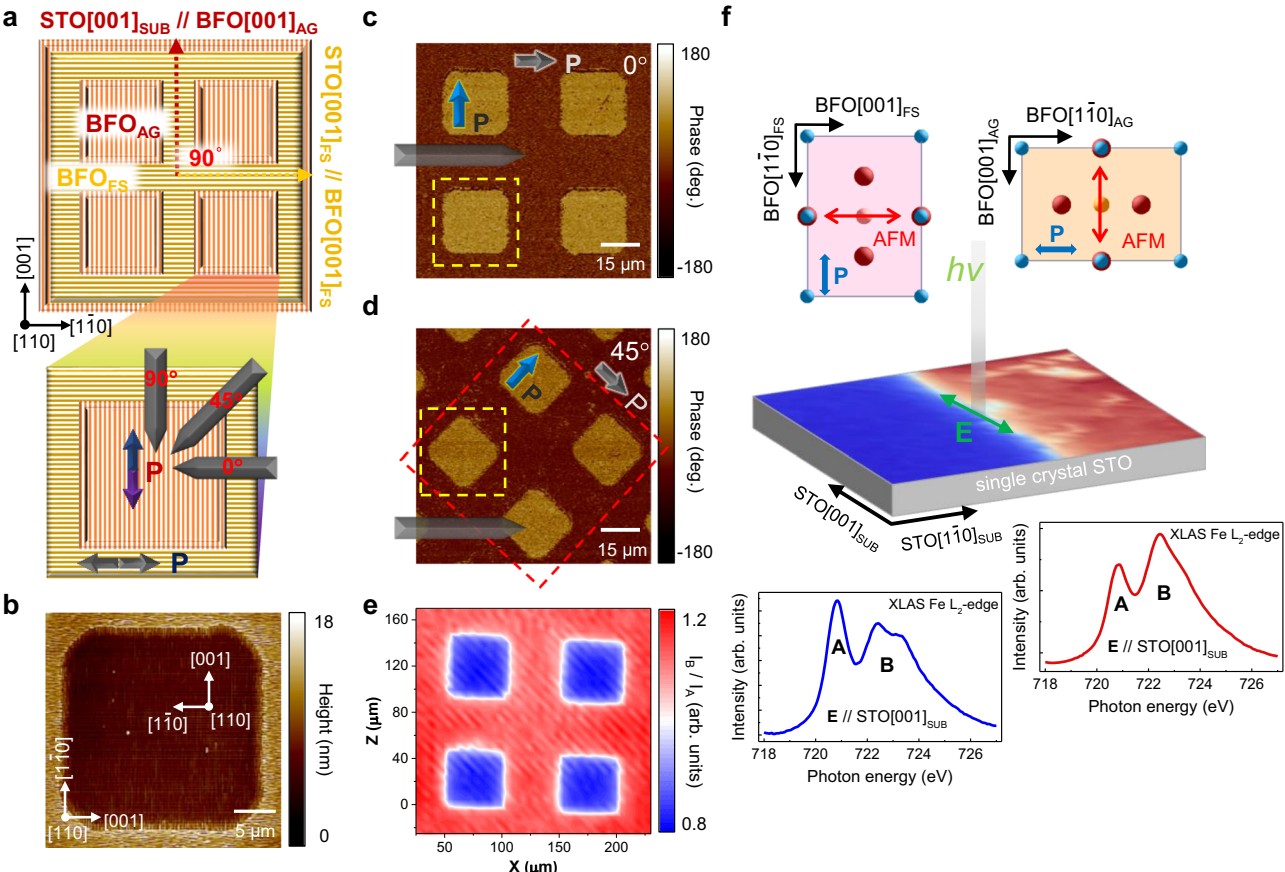

**Fig. 3 Design of patterned lateral homostructures. a** Schematic of the designer BFO (110) lateral homostructures, in which the stripes along BFO [001] direction are designed to be orthogonal. The enlarged area shows the direction of the AFM cantilever. **b** Topography image of the patterned lateral (110)-oriented BFO homostructures. The orthogonal stripes of BFO verify the feasibility of the designer patterns. **c, d** In-plane PFM phase images of the lateral BFO (110) homostructures with the probing cantilever parallel and 45° to [001] direction of FS-STO. The net polarizations of $BFO_{AG}$ and $BFO_{FS}$ are shown in blue and gray arrows, respectively. **e** $I_B/I_A$ ratio of XLAS mapping on the patterned lateral homostructures. **f** $I_B/I_A$ ratio of XLAS mapping across the boundary of $BFO_{AG}$ and $BFO_{FS}$ area. The XLAS Fe $L_2$-edge spectra taken from the as-grown BFO film and the BFO film on FS-STO are presented in blue and red colors, respectively (lower panels). The green arrow shows the electric polarization of the incident X-ray. The upper panels schematically show the correlation of the ferroelectric polarization and antiferromagnetic axes of the patterned homostructures.

The size and the arrangement of the patterned squares are consistent with the PFM results and our designer pattern. The Fe $L_2$-edge XLAS spectra taken from $BFO_{AG}$ film and $BFO_{FS}$ are shown in Fig. 3f, in which a clear dichroism correlated to the relative orientation of polarization vector **E**, crystallography-axis, and the antiferromagnetic axis[27–32] can be observed. To further extract the information provided from the dichroism, we performed configuration interactions cluster calculations. It is a theoretical approach that takes into account the full-multiplet Coulomb interaction, local crystal field, local spin-orbital coupling, local spin direction, and hybridizations with ligand atoms[33–35], and a well-proven tool that can be adopted to effectively simulate the X-ray absorption spectra. In Supplementary Fig. 5, we show the calculated XLAS of both $BFO_{AG}$ to $BFO_{FS}$ regions with **E**//STO $[001]_{SUB}$ based on two different scenarios: (i) only the antiferromagnetic axis rotated by 90 degrees, and (ii) both antiferromagnetic axis and crystallography-axis rotated by 90 degrees. By comparing the experimental and simulated results, two nearly identical dichroism spectra can be obtained when scenario (ii) is adopted. Namely, our simulation indicates that the antiferromagnetic axes of BFO grown on pristine STO and FS-STO regions are perpendicular to each other, as depicted in Fig. 3f. The controllable arrangement of both ferroelectricity and

antiferromagnetic axes using BFO validates the feasibility to design lateral crystalline orientation/domain patterns and related homostructures with conjunction twist angle tunability.

The transition region and interfaces in strongly correlated oxides, for example, domain walls and phase boundaries, always exhibit unconventional physical phenomena and enhanced functionalities, such as low dimensional interfacial conduction, superconductivity, enhanced magnetism and so on[8,36,37]. Here, we further demonstrate the versatility of constructing lateral homostructures composed of the polymorphs with different crystalline symmetry via the insertion of epitaxial freestanding layers. To do so, freestanding (001)-oriented STO layer was transferred onto LaAlO$_3$ (LAO) substrate with ~50% coverage to create a template offering two different lattice mismatches. After deposition, BFO grown on FS-STO exhibits rhombohedral-like phase (R-BFO), while the BFO grown on LAO possesses tetragonal-like phase (T-BFO), as illustrated in Fig. 4a. From the AFM images shown in Fig. 4b, a phase boundary separating two different crystallography of BFO was artificially created. It is worth mentioning that the coexisted yet randomly-distributed T-R phase boundary of BFO can be found in a strain-driven morphotropic mixed-phase system, where the BFO thin films are grown on LAO substrate[9]. As shown the morphology cross-section profile in Fig. 4c, a steep and clear boundary between

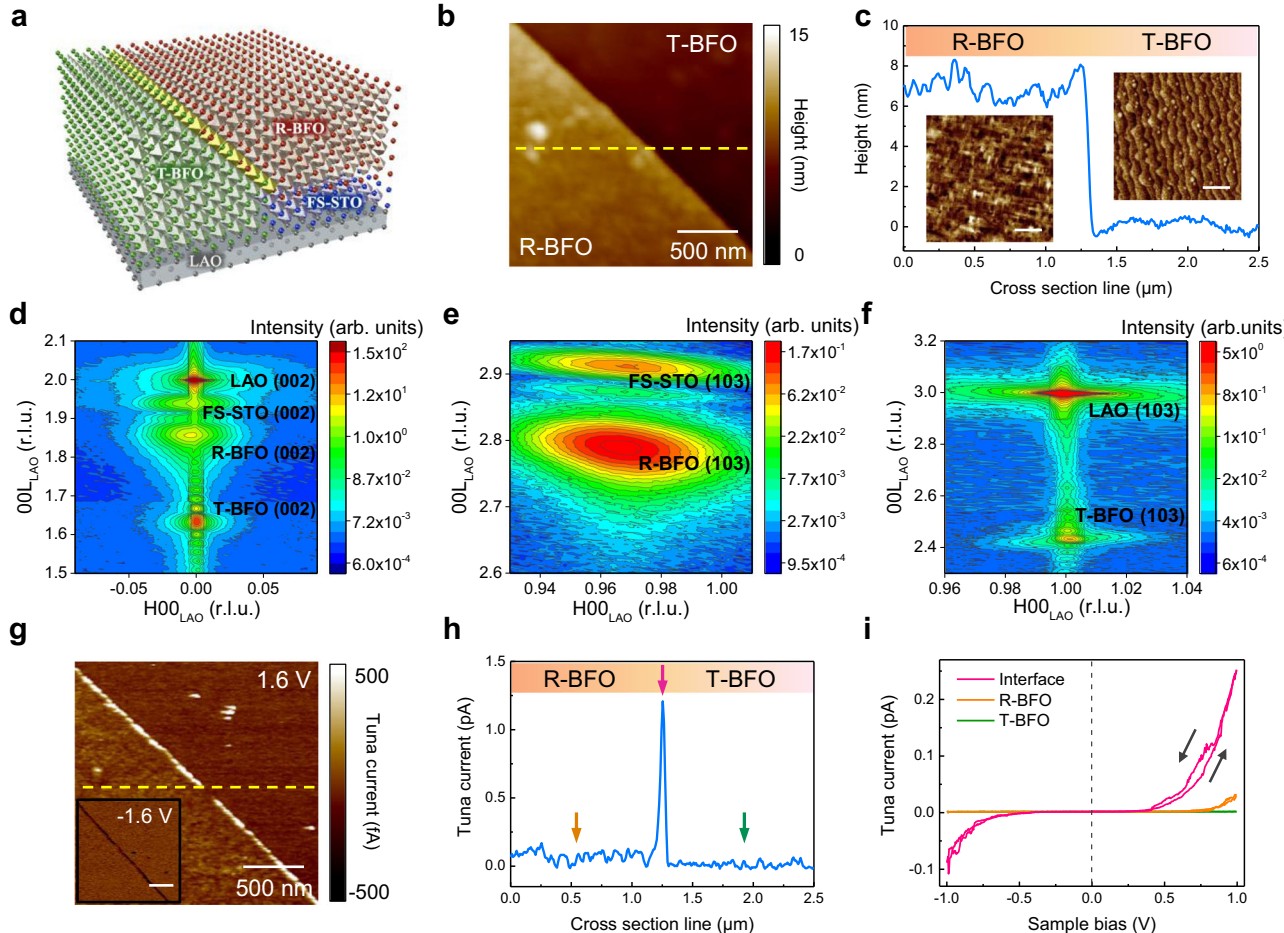

**Fig. 4 Artificial creation of BFO polymorph phase boundary via twisted templates. a** The designer lateral homostructure composed of coexisted tetragonal-like (T-BFO) and rhombohedral-like (R-BFO) BFO phases. Freestanding STO layer (FS-STO) (blue layer) was transferred onto LAO substrate (gray substrate), forming a twisted template providing distinct lattice constrains on both sides. The green and red lattices represent the T-BFO and R-BFO phases, respectively. **b** Surface topography image of the designed lateral homostructure. **c** Cross-section height profile at the dashed yellow line in (**b**) and the enlarged topography images of T-BFO and R-BFO regions. The scale bars refer to 500 nm. **d–f** Corresponding RSM images around (**d**) LAO (002), (**e**) FS-STO (103), and (**f**) LAO (103) taken from T-R phase boundary, R-BFO, and T-BFO regions, respectively. The reciprocal lattice unit (r.l.u) is normalized to the LAO substrate (1 r.l.u. = $2\pi/a_{LAO}$, $a_{LAO} = 3.787$ Å). **g–i** Conductive AFM image (**g**) taken at the phase boundary with a tip bias of 1.6 V, where the left-bottom inset shows the conductive AFM image taken with a tip bias of −1.6 V. (**h**) Corresponding cross-section current profile at the dashed yellow line shown in (**g**). (**i**) Corresponding I–V curves taken at T-R boundary (pink), R-BFO (orange), and T-BFO (green), respectively.

R-BFO and T-BFO can be identified, resulting in a step height ~5 nm, which is consistent with the thickness of FS-STO. The structures of BFO on freestanding STO layer and LAO substrate are then revealed by X-ray diffraction, as shown the RSM mappings in Fig. 4d–f, where the feature peaks of T-BFO, R-BFO, STO and LAO are labeled correspondingly. RSM images near LAO (103) and STO (103) indicate that T-BFO and R-BFO are subject to the epitaxial strain applied by LAO substrate and FS-STO, respectively. Detailed structural properties related to the symmetries of BFO polymorphs, FS-STO and LAO substrate can be found in Supplementary Notes 4 and 5 and Supplementary Figs. 6 and 7.

To probe the local electrical property near the homojunction/boundary between R-BFO and T-BFO, conductive atomic force microscopy (C-AFM) was employed, which simultaneously mapped the surface morphology and corresponding conductivity. The corresponding C-AFM and cross-section images taken with applied tip bias of 1.6 V shown in Fig. 4g, h indicate an obvious enhanced electrical conduction at the T-R phase boundary compared to the individual R-BFO and T-BFO region. The interfacial conduction was further examined by the I–V curves shown in Fig. 4i. The artificial

boundary of the homostructure exhibits a nonlinear Schottky diode-like characteristic with small hysteresis. In such a system, there are several mechanisms that might dominate the potential factors for the modification of electronic structures at the artificially created phase boundary, leading to the enhanced conductivity. First, the ferro-electric and structural discontinuity at the interface is correlated to the local distortion and band-alignment of the homostructure, which might affect the charge accumulation and the electronic band structure across the interface[38,39]. Second, the modification of chemical bonds at the semi-coherent interfaces accompanies the reconstruction of the defect level at the interface. Third, the changes in Fe–O–Fe bond angle and unusual oxygen octahedral rotation at the interface could affect the orbital overlapping and thus varying the bandgap at the phase boundary[40,41]. To this end, the modification of local band structure as well as the accumulation of the charged defects at the phase boundary all contribute to the enhanced conductivity at the artificial R-BFO and T-BFO interface. The artificially created homojunctions with tunable polarization and lattice discontinuity can also be utilized to derive strong second harmonic generation (see Supplementary Note 6 and Supplementary Fig. 8) and to develop high-frequency terahertz (THz) emitters, which are

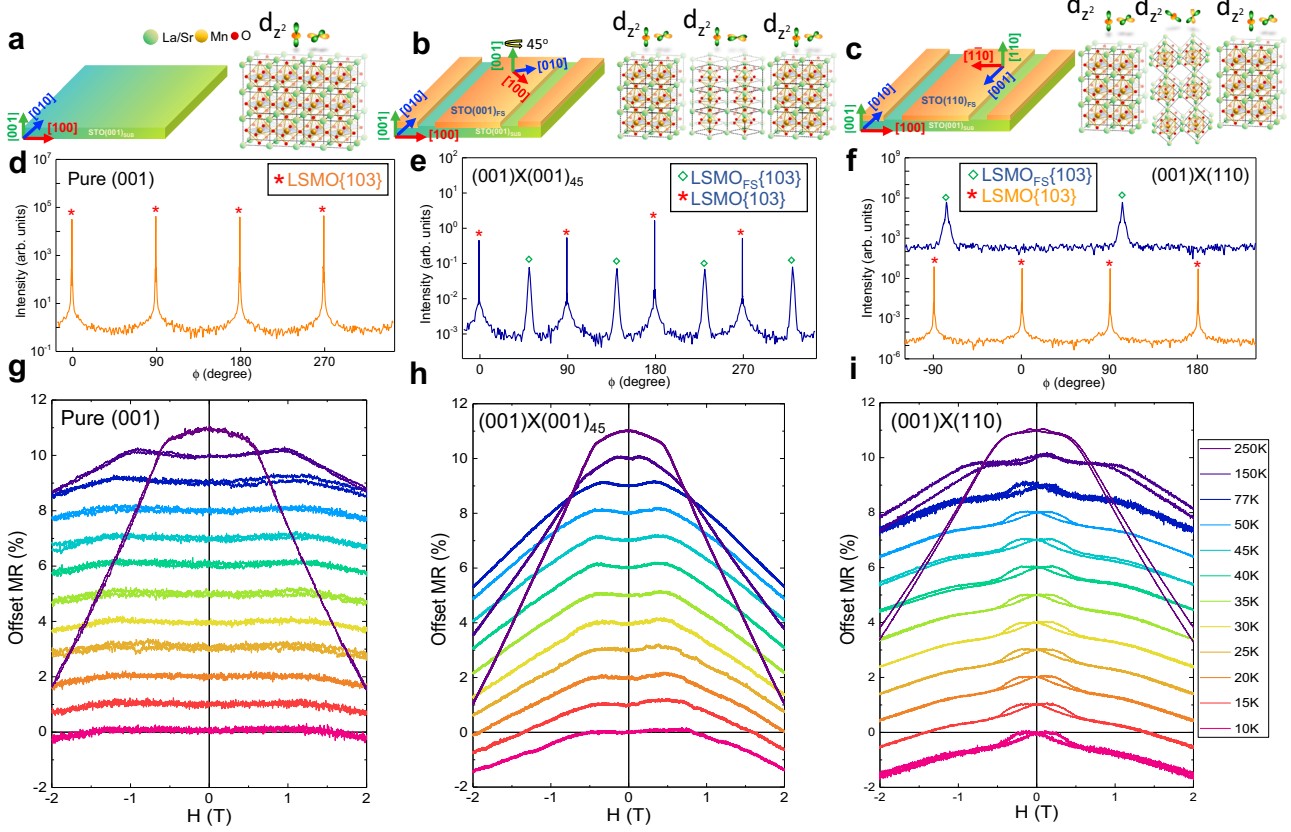

**Fig. 5 Magnetotransport on multijunction LSMO lateral homostructures with alternating orbital configurations. a–c** Schematic illustrations (left) of (**a**) reference (001)-oriented STO substrate, (**b**) twisted array template composed of alternating (001)-oriented STO substrate and FS-STO layers rotated by 45 degrees in-plane, and (**c**) twisted array template composed of alternating (001)-oriented STO substrate and (110)-oriented FS-STO layers. The designed lattice and representative $d_{z^2}$ and $d_{x^2-y^2}$ orbital of LSMO after deposition are illustrated in the right panels. **d–f** Corresponding in-plane X-ray phi scan of {103} reflections of LSMO grown on templates shown in (**a–c**). Temperature-dependent transport measurements of (**g**) pure (001), (**h**) (001) x(001)$_{45}$ LSMO, and (**i**) (001)x(110) LSMO homostructures. The magnetoresistance changes are defined as ($\rho_H - \rho_o/\rho_o$) × 100 (%). The external magnetic field is applied along [001] direction of STO substrates.

the key components toward terabit wireless communication, bio imaging, and astrophysics.

In order to verify our method can be universally adopted to manufacture different complex systems, multijunction twisted lateral (La, Sr)MnO$_3$ (LSMO) homostructures were fabricated. LSMO is a classic half-metallic oxide that exhibits large spin polarization and colossal magnetoresistance (CMR) effect[42]. The magnetotransport properties are closely correlated to strain states and grain boundaries of LSMO heterostructures. To date, the lateral boundary control of LSMO has rarely been systematically studied. Herein, three types of LSMO lateral homostructures with different alternating lattice arrays were fabricated, using twisted templates illustrated in left panels of Fig. 5a–c. As a reference sample, the X-ray phi scan of (001)-oriented LSMO grown on STO substrate (template shown in Fig. 5a) indicates high-quality thin film with four-fold symmetry (Fig. 5d). The second LSMO homostructure was grown on twisted template shown in Fig. 5b, in which two sets of four-fold LSMO {103} peaks can be revealed by XRD phi scan (Fig. 5e), indicating that the (001)-oriented LSMO$_{FS}$ is rotated 45° in the in-plane direction. To ease relevant discussion later, the second sample is denoted as (001)x(001)$_{45}$ LSMO homostructure. Following a similar concept, the third twisted system with alternating (001)- and (110)-oriented LSMO, denoted as (001)x(110) LSMO homostructure, was fabricated by growing LSMO on the twisted template shown in Fig. 5c. The XRD phi scan is shown in Fig. 5f, once again, supports the suspected crystal geometry.

Given that the electron orbitals rotate accordingly with the rotated crystal lattices, the patterned single crystal arrays fabricated via our approach effectively enable the orbital control in a well-defined alternating manner. The representative $d_{z^2}$ and $d_{x^2-y^2}$ orbitals of Mn ions of pure LSMO, (001)×(001)$_{45}$ LSMO, and (001)×(110) LSMO are schematically illustrated in right panels of Fig. 5a–c, respectively. To reveal how the artificially created alternating orbital configurations could alter the intrinsic properties of LSMO, magnetotransport measurements were carried out. Figure 5g–i shows the magnetoresistance (MR), defined as ($\rho_H - \rho_o/\rho_o$) $\Delta\rho(H)/\rho(0)$, of the three aforementioned systems at different temperatures (see Supplementary Fig. 9 for the measurement details). The pure LSMO thin film (Fig. 5g) shows a negligibly small negative MR at low temperature and it increases with increasing temperature, a phenomenon expected for conventional LSMO heterostructures. In contrast, the low-temperature MRs in (001)×(001)$_{45}$ LSMO and (001)×(110) LSMO homostructures are significantly larger compared to that in pure LSMO sample. Previous studies indicated that structural disorders such as phase or grand boundaries presented in manganite films would create additional energy barrier to the spin-polarized tunneling process, resulting in the enhanced low-field MR phenomenon[43–46], which is in nice agreement with our observations. Additionally, it is remarkable that an unconventional butterfly-shaped hysteresis MR is observed in (001)×(110) LSMO, as compared with pure LSMO and (001)×(001)$_{45}$ LSMO. This is due to the fact that the (001)×(110) LSMO is designed to exhibit alternating $d_{z^2}$ orbital rotations with respect to the applied magnetic field, whereas the pure LSMO and

$(001)\times(001)_{45}$ LSMO samples do not (see Fig. 5a–c). In such a scenario, the spin-polarized electrons experience an artificially created magnetic inhomogeneity under external magnetic field, resulting in the hysteresis MR loops. These results successfully demonstrate the construction of alternating single-crystal arrays as well as controllable orbital arrangements of LSMO, validating the generic capability to fabricate twisted lateral complex systems using our approach.

Through the insertion of the extremely thin freestanding layers, we are able to synthesize twisted lateral homostructures with crystal orientation, epitaxy constrain and phase conjunction controllability. The unbounded combination based on the proposed method is not limited to the relative twist of the same lattice orientation, the combination with different out-of-plane lattice orientations is also feasible (also refer to Supplementary Note 7 and Supplementary Fig. 10). The controllable twist angle at the boundary of the same materials offers a unique platform for developing emergent phenomena. Such an approach is compatible with the modern photolithography process, through which we have successfully demonstrated the patterned lateral homostructures with artificially designable ferroic orders and orbital configurations. By transferring freestanding thin film onto the substrate with dissimilar lattice structure, we further show that distinct phases of the same material can be simultaneously assembled, forming a sharp interface exhibiting distinct physical properties. Furthermore, we also reveal that the magnetotransport properties of twisted systems can be artificially manipulated via designed electron orbital configurations. Our results lay a groundwork for the design of twisted lateral homostructures, which might contribute to advanced manipulation of strongly correlated phenomena in complex systems through artificially modified electronic structure and crystal geometry.

## Methods

**Synthesis of the lateral homostructures**. The samples were fabricated by pulsed laser deposition (PLD), using KrF (248 nm) excimer laser. The $SrTiO_3$ (STO) thin films were deposited on $(La, Sr)MnO_3$ (LSMO) buffered (110)-oriented STO substrates. The STO was deposited at oxygen pressure of 100 mTorr at 700 °C with a laser power of 250 mJ and laser repetition rate of 10 Hz, while the LSMO was deposited at oxygen pressure of 100 mTorr at 750 °C with a laser power of 250 mJ and laser repetition rate of 10 Hz. Thereafter, the heterostructure was immersed in hydrochloric acid to dissolve LSMO buffered layer and to separate STO film from single-crystal substrate. The freestanding STO was then transferred onto another (110)-oriented STO single crystal substrate, having a twist angle with respect to the STO single crystal substrate. Last, the $BiFeO_3$ thin film was deposited at oxygen pressure of 85 mTorr at 680 °C with a laser power of 250 mJ and laser repetition rate of 10 Hz.

**Structural analysis**. The crystal structure of the as-grown thin films and homostructures was characterized by synchrotron-based X-ray high-resolution 8 circle diffractometer with beam energy of 10 keV and size of $0.3 \times 0.7$ mm$^2$ at beamlines TPS-09A, TLS-17B and TLS-13A1 in the National Synchrotron Radiation Research Center, Taiwan.

**Transmission electron microscopy**. Cross-sectional TEM specimens were prepared by focused ion beam (FIB, Helios G4UX, FEI, America). In order to protect the thin film, a ~50 nm thick C layer was coated on the surface of the film before the FIB milling. HRTEM observations were carried out using a field emission microscope (JEM-2100F, JEOL, Japan) at an acceleration voltage of 200 kV. Further HAADF imaging, EDS mapping, and EELS analyses were performed with an Cs-corrected STEM operating at 300 kV (Grand ARM-300, JEOL, Japan) equipped with an X-ray energy dispersive spectrometer (JED-2300T) and Gatan image filter (GIF) system. In EELS analyses, an entrance aperture of 2.5 mm was used and the energy resolution of 1.0 eV was determined by measuring the full-width half-maximum (FWHM) of the zero-loss peak with the energy dispersion of 0.25 eV/channel. The EELS spectra were recorded with spectrum image mode to increase the signal/noise ratio, and the acquisition time was about 0.2 s per pixel (total acquisition time is about 3 min).

**Lateral homostructure devices**. The lateral homostructure devices were fabricated by a combination of photolithography and etching processes. First, freestanding STO (110) thin film was transferred onto (110)-oriented STO substrate with 90 degrees twist angle. Later, photoresist with geometric patterns was created on the twisted FS-STO/STO template via the photolithography method. The patterns were then covered

by 200 nm chromium, deposited by an e-beam evaporator, as a protective layer. After the removal of photoresist, the revealed FS-STO (squares) were further etched by the reactive ion etching (RIE). During the etching process, the chamber was kept at 100 mTorr with a methane flow of 50 sccm and the plasma power generator was controlled at 500 mW where the etching rate is 0.4 nm/min. The well-defined pattern of FS-STO/STO can then be obtained after the removal of the chromium layer by CR-7 chromium etchant. Thereafter, BFO was grown on the patterned FS-STO/STO template using PLD.

**X-ray absorption spectroscopy**. The X-ray absorption spectroscopy measurements were conducted at TPS-45 beamline, National Synchrotron Radiation Research Center (NSRRC). The beam resolution was set to be around 50 meV with the spot size around 3*3um.

**Configuration interactions cluster calculations**. The calculation was processed using Quanty[47–50], a script language which allows the user to program and resolve many bodies quantum mechanical problems in second quantization. Quanty-based configuration interactions cluster calculations are then utilized to simulate $L_{2,3}$-edge X-ray absorption spectra of the 3d-transition metal oxides[47].

**Atomic force microscopy and piezo-force microscopy**. The SPM-based studies (atomic force microscopy and piezo-force microscopy) for topography and PFM were operated via the closed-loop commercial scanning probe microscope system (Dimension Icon, Bruker) with software Nanoscope 9.14 in scanasyst mode and vertical-optimized PFM mode, respectively. All PFM images were recorded in the matrix of $256 \times 256$ with an a.c. voltage of 2 V and 7 kHz applied on the Pt/Ir-coating conductive probe (elastic force constant ~7.4 N m$^{-1}$). The in-plane PFM images of BFO homostructures with different twist angles were obtained using a lock-in amplifier (SR-830, Stanford Research Systems).

**Second harmonic generation measurements**. The SHG measurements were performed by a Ti:sapphire laser with a center wavelength of 800 nm, a repetition rate of 5.2 MHz, and a pulses duration of 60 fs. A half-wave plate was placed beyond the excitation laser. We rotated the half-wave plate to control the polarization angles $\phi$ of the excitation laser beam. A polarization analyzer was mounted to analyze the p-polarized and s-polarized components of the SHG. The SHG mappings were performed on a home-built laser scanning confocal microscope. A Nikon objective lens (×10; NA = 0.3) was used to focus the laser light onto the BFO thin film with a spatial resolution of ~4 μm.

**Magnetotransport measurements**. The devices with Hall bar geometry were measured using the standard four-terminal lock-in techniques, carried out in a cryogenic system with a superconducting magnet. The longitudinal voltage was measured by a lock-in amplifier with an a.c. excitation current of 500 nA at the frequency $f = 77$ Hz being driven through the devices to obtain the magnetoresistance.

## Data availability
The data supporting the key findings of this study can be found within the article and its Supplementary Information files or are available from the corresponding author upon request. Source data are provided with this paper.

## Code availability
The code that was used to analyze the experimental data is available from the corresponding authors upon request.

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

## Acknowledgements

This work is supported by the Ministry of Science and Technology (MOST) in Taiwan under grant no. MOST 110-2636-M-006-003 (Young Scholar Fellowship Program) and 110-2628-M-006 -002-MY3 (J.-C.Y. and Y.-C.C.). R.H. acknowledges support by the National Key Research and Development Program of China (2017YFA0303403) and the National Natural Science Foundation of China (Grant No. 61974042). C.-F.C. and C.-Y.K. acknowledge support from the Max Planck-POSTECH-Hsinchu Center for Complex Phase Materials. The authors thank MSSORPS Co., Ltd. for high-quality TEM sample preparation and preliminary observation of HRTEM. The authors thank Ms. Jo-Chi Hu and Mr. Kei-Jie You for preliminary sample preparation. The research is also supported in part by Higher Education Sprout Project, Ministry of Education to the Headquarters of University Advancement at National Cheng Kung University (NCKU).

## Author contributions

P.-C.W., Y.-C.L., and C.-C.W. processed the sample growth and X-ray diffraction analysis. S.-Z.H., Y.-D.L., and Y.-C.C. conducted characterizations of scanning probe microscopy, and analyzed the data. Q.Z. and R.H. carried out transmission electron microscopy characterization, processed high-angle annular dark-field imaging and resolved the crystal structure. J.Z and C.-G.D. conducted the first-principle calculations. C.-C.C., C.-F.C., and C.-Y.K. processed X-ray absorption-based measurements and analyzed the data. W.-Y.T., C.-M.T., and C.-W.L. carried out second harmonic generation measurements. K.-E.C, Y.-W.C., and T.-M.C. conducted magnetotransport measurements. J.-C.Y. led the project, conceived main idea, and co-wrote the manuscript draft with P.-C.W. All authors contributed to the manuscript.

## Competing interests

The authors declare no competing interests.
