## [Peer Review File · Nature Communications]

Title: Twisted oxide lateral homostructures with conjunction tunabilityREVIEWER COMMENTS

Reviewer #1 (Remarks to the Author):

The current manuscript describes a novel process to create twisted structures made by epitaxially grown oxides. The oxides are grown epitaxially with a sacrificial buffer layer and then they are released and, as a free-standing layer, they are deposited at an angle (twisted) with respect to the substrate materials. Afterwards a new layer is epitaxially deposited on top the twisted structure that displays two very different structures in the twisted and untwisted areas.

The paper will arise significant interest from the electronics oxide community and I believe it deserve publication in a high impact journal such as Nat. Comm. after some points have been addressed.

Specifically:

- 1) the expression used "multiple conjunction degree of freedom" is very unclear and difficult to grasp. The authors should think of a different way to bring that aspect in.
- 2) The figure captions are not very helpful to follow the figures. In figure 1, the numbers indicating the stages of the process are missing in the figure. In figure 4a) the colors are wrong.

Reviewer #2 (Remarks to the Author):

This manuscript, "Twisted oxide lateral homostructures with conjunction tunability" by Ping-Chun Wu et al. describes a method to achieve an oxide film consisting of two different crystallographic orientations that are patterned in a user-defined way. Not only can the user choose the out-of-plane orientation of the two orientation variants, but the user can define the in-plane rotation between these variants and even pattern the extent of each variant and the geometry of the in-plane boundaries between the variants. This is nice work, but it fails to refer to the closely related work that has already been done. When the existing work is recognized, the apparent novelty of the present work vanishes. For this reason, in my opinion this work does not meet the standards for publication in Nature Communications.

To begin with, the authors describe a means of using an epitaxial (La,Sr)MnO₃ sacrificial layer to achieve free-standing layers of SrTiO₃ that can be transferred onto the bare surface of a SrTiO₃ substrate with user-controlled positioning. Though not referred to, this method is essentially identical to that used by Bakaul, S. R. et al. Single crystal functional oxides on silicon. Nature Communications 7, 1083 (2016). This 2016 paper uses (La,Sr)MnO₃ as a soluble buffer layer to release epitaxial oxide films from underlying oxide single crystal substrates, including SrTiO₃. The 2016 work then transfers the oxide layers onto a silicon wafer, but conceptually they can be transferred onto any substrate.

Considerable prior work has been done to utilize epitaxy, in manners completely analogous to what is described in this manuscript, to provide overlying films with two (or more) orientations controlled by epitaxy, where the boundaries between the overlying epitaxial grains are controlled in-plane in both lateral position with full control of the grain boundaries. Two of these methods (bicrystals, tricrystals,

and bi-epitaxy) are described in Ref. 34 of the current manuscript, but not described in sufficient detail to enable the reader to recognize how what is achieved in the current manuscript does not go far beyond what has already been achieved. The use of bicrystals and tricrystals (see Fig. 2 of Ref. 34) involves sintering together two or more single crystals with controlled angles between the crystals so that epitaxy can be used to achieve controlled grain boundaries (tilt and twist boundaries) of controlled geometries at a controlled location. This technique is at least 50 years old. See Schober T. & Balluffi, R. W. Dislocation sub-boundary arrays in oriented thin-film bicrystals of gold. *Philos. Mag. A* 20, 511–518 (1969). Further, the epitaxial technique known as “bi-epitaxy” (see Fig. 7 of Ref. 34) was developed in the 1990s so that arbitrarily many in-plane epitaxial junctions at lithographically defined locations with user-controlled geometry could be achieved to make superconducting Josephson junctions. This technique is described in the following references and related patents:

Char, K., Colclough, M. S., Garrison, S. M., Newman, N. & Zaharchuk, G. Bi-epitaxial grain boundary junctions in YBa₂Cu₃O₇. *Appl Phys Lett* 59, 733–735 (1991).

Char, K., Colclough, M. S., Lee, L. P. & Zaharchuk, G. Extension of the bi-epitaxial Josephson junction process to various substrates. *Appl. Phys. Lett.* 59, 2177–2179 (1991).

Wu, X. D., Luo, L., Muenchausen, R. E., Springer, K. N. & Foltyn, S. Creation of 45° grain-boundary junctions by lattice engineering. *Appl. Phys. Lett.* 60, 1381–1383 (1992).

Another example related to the current manuscript is the hybrid orientation technology (HOT) developed by IBM starting in 2003. Here using wafer bonding between (100)- and (110)-oriented silicon wafers followed by epitaxy, regions of these two desired orientations in which high mobility n-channel and p-channel transistors can be grown are made side-by-side with full lithographic control. A nice overview of this technique is given by Yang, M. et al. Hybrid-Orientation Technology (HOT): Opportunities and Challenges. *IEEE Trans. Electron Dev.* 53, 965–978 (2006).

In conclusion, this manuscript reports nice work, but in my opinion it lacks sufficient novelty to warrant publication in *Nature Communications*. I also find it concerning and sloppy of the authors to not properly credit those who have done closely related and relevant work prior to them.

Reviewer #3 (Remarks to the Author):

The conventional epitaxial growth of thin films can only control the structure in film thickness direction. In this work, the authors propose a fabrication approach to manipulate the lateral structure by fabricating and twisting free standing substrate film relative to the original substrate, and growing films on the free-standing film and the original substrate. The authors provide detailed characterization results to demonstrate the feasibility of the approach to obtain lateral homostructures with different twisted angles and in different types of materials. The work is interesting, but I have several questions regarding their results:

(1) Since in some cases, the misalignment between BFO_{AG} and BFO_{FS} is large, how does the interface accommodate the large alignment? The authors observe the existence of oxygen

vacancy in the interfacial region, but oxygen vacancy alone may not accommodate the lattice mismatch between the two parts of the film. Or the two parts of the film just form the physical interface without considering the misalignment. I hope the authors could provide the detailed explanation about the interfacial microstructure.

(2) The intention of this work is to control the lateral structure of epitaxial film so that some unusual physical properties can be achieved. The authors demonstrate that the polarization and antiferromagnetism of BFO films can be controlled by their proposed fabrication approach, but they didn't show the uniqueness of the approach to obtain unusual multiferroic properties. The nonlinear conductivity is also presented, but the result is also not unique. The demonstration of the effectiveness to achieve unusual physical properties by using this fabrication approach is highly welcomed and this can strengthen the importance of this work.

(3) Other minor issues:

The I-V curve in Fig. 4 could be presented separately to demonstrate the physical outcome of structural manipulation.

The twisted angle should be indicated in the figure caption of the Fig. 2 and Fig. S4.

Summary of the revisions:

1. Fig. 1a and Fig. 4a are revised for better understanding according to Referee 1's reminders.
2. Introduction parts are rephrased to highlight existing methods for fabricating homostructures and the progress we would like to achieve through our work.
3. As suggested by Referee 3, further TEM analysis of the microstructure of $\text{BFO}_{\text{AG}}\text{-BFO}_{\text{FS}}$ interface is incorporated into the main text as well as the revised Fig. 2 of this revision.
4. Second harmonic generation (SHG) measurements with confocal microscopy mapping have been performed to reveal the well-controlled crystalline symmetry and the unconventional increase in nonlinear optical coefficient at lateral homojunction/boundary (Fig. S8).
5. Original Fig. 5 (YBCO homostructures) and associated paragraphs are removed.
6. Additional electron orbital configuration control and multijunction magnetotransport measurements based on twisted LSMO homostructures are incorporated into the revised manuscript (new Fig. 5).
7. Observations of enhanced magnetoresistance across periodically alternating lattice arrays are included in new Fig. 5.
8. The presentation of Fig. 4 is revised based on Referee 3's suggestions.
9. Typos are corrected in the revised manuscript.

Response to Referee #1

Reviewer comments:

The current manuscript describes a novel process to create twisted structures made by epitaxially grown oxides. The oxides are grown epitaxially with a sacrificial buffer layer and then they are released and, as a free-standing layer, they are deposited at an angle (twisted) with respect to the substrate materials. Afterwards a new layer is epitaxially deposited on top the twisted structure that displays two very different structures in the twisted and untwisted areas.

The paper will arise significant interest from the electronics oxide community and I believe it deserve publication in a high impact journal such as Nat. Comm. after some points have been addressed.

Response:

We sincerely thank the referee for the positive comments and the recommendation on our manuscript. We have further revised our manuscript based on his/her remarks. All the revisions and responses are detailed in the following section.

Reviewer comments:

1) The expression used "*multiple conjunction degree of freedom*" is very unclear and difficult to grasp. The authors should think of a different way to bring that aspect in.

Response:

We're sorry about the imprecise expression in previous manuscript. To better convey the concept and advantage of this work, we have rephrased the sentence as follows:

"In this work, we reveal the growth of twisted oxide lateral homostructure with controllable in-plane conjunctions."

Reviewer comments:

2) The figure captions are not very helpful to follow the figures. In figure 1, the numbers indicating the stages of the process are missing in the figure. In figure 4a) the colors are wrong.

Response:

We are sorry for the confusion and mistake we made in previous Fig. 1a and Fig. 4a and we thank the kind reminder from the referee. The missing numberings in Fig. 1a and the misplaced color descriptions in the caption of Fig. 4a are corrected in our revised version.

Revised Fig. 1a

Fig. 1 Synthesis of the lateral oxide homostructures. **a** Process flow for fabricating twisted lateral oxide homostructures. (i) Growth of STO with a LSMO sacrificial layer on (110)-oriented STO single crystal substrate. (ii) Etching of the LSMO sacrificial layer. (iii) Releasing and (iv) transferring the freestanding STO layer with designated twisted angle with respect to the selected single crystal carrier substrates. (v) Deposition of BFO thin film on the twisted template. **b** Topography images of (110)-oriented BFO grown on pristine STO, boundary, and FS-STO region, respectively. The $[001]_{pc}$ -oriented stripes serve as the direction reference of epitaxial BFO, identifying the in-plane directions of (110)-oriented BFO and the twisted angle (denoted as angle φ) at the boundary of pristine STO and FS-STO region.

Revised caption of Fig. 4a

Fig. 4 Artificial creation of BFO polymorph phase boundary. *a* The designer lateral homostructure composed of coexisted tetragonal-like and rhombohedral-like BFO phases. Freestanding STO (FS-STO) layer (blue layer) was transferred onto LAO substrate (grey substrate), forming a twisted template providing distinct lattice constrains on both sides. The green and red lattices represent the tetragonal-like (T-BFO) and rhombohedral-like (R-BFO) BFO phases, respectively.

Response to Referee #2

Reviewer comments:

This manuscript, “Twisted oxide lateral homostructures with conjunction tunability” by Ping-Chun Wu *et al.* describes a method to achieve an oxide film consisting of two different crystallographic orientations that are patterned in a user-defined way. Not only can the user choose the out-of-plane orientation of the two orientation variants, but the user can define the in-plane rotation between these variants and even pattern the extent of each variant and the geometry of the in-plane boundaries between the variants. This is nice work, but it fails to refer to the closely related work that has already been done. When the existing work is recognized, the apparent novelty of the present work vanishes. For this reason, in my opinion this work does not meet the standards for publication in *Nature Communications*.

Response:

We sincerely thank the criticism raised by the referee, pointing out the deficiency of our work. In this revised version, related works on the fabrication of epitaxial lateral homostructures are discussed in the introduction parts. To highlight the novelty of our work, we provide a comprehensive comparison between previously established methods and ours, without detracting from the merits of previous works. Furthermore, to highlight the advantages compared to previously established methods, we have exploited more experiments (second harmonica generation and magnetotransport) to reveal the unique capabilities and versatility offered by our approach. Detailed changes and responses are addressed in the following paragraphs.

Reviewer comments:

To begin with, the authors describe a means of using an epitaxial (La,Sr)MnO₃ sacrificial layer to achieve free-standing layers of SrTiO₃ that can be transferred onto the bare surface of a SrTiO₃ substrate with user-controlled positioning. Though not referred to, this method is essentially identical to that used by Bakaul, S. R. et al. Single crystal functional oxides on silicon. *Nature Communications* 7, 1083 (2016). This 2016 paper uses (La,Sr)MnO₃ as a soluble buffer layer to release epitaxial oxide films from underlying oxide single crystal substrates, including SrTiO₃. The 2016 work then transfers the oxide layers onto a silicon wafer, but conceptually they can be transferred onto any substrate.

Response:

We agree with the comments raised by the referee. The freestanding (FS) method by using a soluble (La,Sr)MnO₃ sacrificial layer has been proposed in several important literatures such as Bakaul, S. R. *et al. Nat. Commun.* 7, 10547 (2016)¹ and Shen, L. *et al. Adv. Mater.* 29, 1702411 (2017)², *etc.* Not limited by LSMO, recently, functional oxide films can also be released by adopting a water-soluble Sr₃Al₂O₆ layer^{3,4} or YBa₂Cu₃O_{7-x} sacrificial layer⁵. These works mainly focus on the physical

properties of freestanding (FS) films on certain substrates or the vertically van der Waals stacking structures that cannot be obtained via conventional epitaxial growth route. In contrast, our manuscript combines the well-established FS method with subsequent epitaxial growth, revealing that twisted lateral complex systems can be efficiently fabricated. Taking advantage that the conjunction angles between FS layers and underlying substrate can be well-controlled, we show that the twisted oxide lateral homostructures based on strongly correlated systems can be fabricated to exhibit unconventional physical properties. We never intended to define ourselves as the pioneers of freestanding technique based on sacrificial layers and we apologize for any misunderstandings in our previous manuscript. Also, we're terribly sorry for not properly citing the literatures regarding freestanding processes in previous version. We sincerely thank the kind reminder from the referee, and we have included the aforementioned works in this revision. The revised paragraph is shown below and more discussion can be found in the main text:

"The STO/LSMO/STO heterostructure was then immersed in an acid solution, which was used to dissolve the LSMO layer, separating the grown STO thin film and STO substrate. Similar freestanding process via utilizing LSMO as a sacrificial layer has been reported by Bakaul, S. R. et al. (2016)¹ and Shen, L. et al. (2017)². The freestanding (110)-oriented STO layer (FS-STO) was then obtained and transferred onto the other (110)-oriented STO substrate."

Reviewer comments:

Considerable prior work has been done to utilize epitaxy, in manners completely analogous to what is described in this manuscript, to provide overlying films with two (or more) orientations controlled by epitaxy, where the boundaries between the overlying epitaxial grains are controlled in-plane in both lateral position with full control of the grain boundaries. Two of these methods (bicrystals, tricrystals, and bi-epitaxy) are described in Ref. 34 of the current manuscript, but not described in sufficient detail to enable the reader to recognize how what is achieved in the current manuscript does not go far beyond what has already been achieved. The use of bicrystals and tricrystals (see Fig. 2 of Ref. 34) involves sintering together two or more single crystals with controlled angles between the crystals so that epitaxy can be used to achieve controlled grain boundaries (tilt and twist boundaries) of controlled geometries at a controlled location. This technique is at least 50 years old. See Schober T. & Balluffi, R. W. Dislocation sub-boundary arrays in oriented thin-film bicrystals of gold. *Philos. Mag. A* 20, 511–518 (1969). Further, the epitaxial technique known as “bi-epitaxy” (see Fig. 7 of Ref. 34) was developed in the 1990s so that arbitrarily many in-plane epitaxial junctions at lithographically defined locations with user-controlled geometry could be achieved to make superconducting Josephson junctions. This technique is described in the following references and related patents:

Char, K., Colclough, M. S., Garrison, S. M., Newman, N. & Zaharchuk, G. Bi-epitaxial grain boundary junctions in $\text{YBa}_2\text{Cu}_3\text{O}_7$. *Appl. Phys. Lett.* **59**, 733–735 (1991).

Char, K., Colclough, M. S., Lee, L. P. & Zaharchuk, G. Extension of the bi-epitaxial Josephson junction process to various substrates. *Appl. Phys. Lett.* **59**, 2177–2179 (1991).

Wu, X. D., Luo, L., Muenchausen, R. E., Springer, K. N. & Foltyn, S. Creation of 45° grain-boundary

junctions by lattice engineering. *Appl. Phys. Lett.* **60**, 1381–1383 (1992).

Response:

We sincerely thank the referee for pointing out our deficiency and offering key hints to improve our work. The referee is correct, bi-crystal and bi-epitaxy approaches that allow epitaxial growth of films with different crystal orientations as well as user-controlled boundary positioning have been proposed for decades. In addition to bi-crystal and bi-epitaxy approaches, step-edge method has also been evaluated in this response. Aforementioned approaches that have been adopted for fabricating epitaxial twisted lateral structures are schematically illustrated in following figure (**Fig. R1**⁶). For better comparison, fabrication of twisted lateral homostructures proposed in this work is also schematically shown in **Fig. R2**.

Fig. R1 Illustrations of a bi(tri)crystal method. b, c biepitaxial method. d step-edge approach. Reproduced from Ref⁶.

Arbitrary conjunction angle

Fig. R2 Schematic of twisted oxide lateral homostructures with conjunction tunability (epitaxy with arbitrary misorientation angle) in this work.

The bi(tri)crystal method is fulfilled by epitaxially growing thin films on bi(tri)crystal substrates. As mentioned by the referee, this method involves sintering two or more polished single crystals with controlled angles between the crystals so that epitaxy can be used to achieve controlled grain boundaries (tilt and twist boundaries) in a well-controlled geometries at a desired location^{7,8} (see **Fig. R1a**). Adoption of bi-crystal method has opened an important realm for exploring non-trivial Josephson junctions based on high-temperature superconductors such as YBCO, setting a milestone in modern physics research. Nevertheless, using our approach, we hope to complement the shortcomings limited by the necessary high-temperature annealing process required by bi-crystal method. When it comes to preparing a twisted template consisting of two different substrates, which can be used to offer distinct lattice constrains on both sides of the boundary (ex. twisted $\text{LaAlO}_3/\text{SrTiO}_3$ bi-crystal), high-temperature annealing process would cause severe interdiffusion at the boundary. This is the main consideration to demonstrate lateral homostructures composed of the polymorphs stabilized by distinct strains in our work (*for twisted BFO homostructure on $\text{LaAlO}_3/\text{SrTiO}_3$ template, please refer to Fig. 4 and relevant paragraphs in the main text*).

Additionally, it is difficult to prepare crystal templates with multiple junctions composed of small and repetitively alternating segments with different orientations based on bi-crystal method. To demonstrate that our approach can complement this disadvantage, LSMO lateral homostructures exhibiting repetitively alternating lattice arrays are investigated in this revision (*please refer to revised Fig. 5 and relevant paragraphs*). Through cutting-edge lithography process, we could further define scalable patterns down to tens of nano-meters, which is hard to be achieved using bi-crystal method.

The bi-epitaxy method relies on epitaxial growth atop (patterned) oxide buffered layers, as schematically shown in **Fig. R1b&c**. The buffer layer plays a focal role to accommodate lattice mismatch and free energy difference between the substrate and the grown thin films. In most of cases, the misorientation angle is confined to 45 degree⁶, forming 45° [001]-twisted boundaries^{9,10,11}. In this regard, an arbitrary misorientation angle control is hard to be achieved by using bi-epitaxy method. Step-edge approach, direct epitaxial thin film growth on substrates with ion beam etched steps on the surface, shares similar drawback as bi-epitaxy method. The artificially-created misorientation is shown to be dependent on the etched depth and step slope. However, to the best of our knowledge, only misorientation angles of 45° or 90° can be reproducibly demonstrated^{12,13}. To offer a comprehensive comparison, Table R1 summarizes the pros and cons of the approach proposed in our work and previously established methods including bi(tri)crystalline, biepitaxial and step-edge approaches.

Table R1 Comparison of techniques toward the fabrication of epitaxial lateral homostructures

Key features \ Methods	bi(tri)crystalline	biepitaxial	step-edge	our approach
Controllable misorientation angle at lateral interfaces (Arbitrary twist angle)	✓	×	×	✓
Controllable location of twisted interfaces on carrier substrates	✓	✓	✓	✓
Nanometer step height	✓	✓	×	✓
Capability to fabricate multiple junctions/homointerfaces with small size and high density	×	✓	✓	✓
Offering different strains/substrates on both sides of the interfaces	×	✓	×	✓

To shortly summarize the progress based on our approach, some main advantages are considered:

- (1) Multiple misorientation boundaries (multi-junctions) with controllable periodicity down to tens of nanometers can be fabricated.
- (2) Misorientation angles could be properly controlled by rotating freestanding layers, as compared to limited twisted angles using bi-epitaxy and step-edge approaches.
- (3) Twisted templates composed of different elements can be realized, thus different strain states can be generated by different lattices on both sides.

As emphasized in previous responses, we don't have any intention of overclaiming the advantages using our approach. Nevertheless, the existing methods for fabricating twisted homostructures do not

simultaneously possess all the above-mentioned merits. We sincerely hope the approach developed in this work can add additional degrees of freedom when designing epitaxial lateral homostructures. Also, we sincerely apologize for our mistake without mentioning detailed information regarding existing methods such as bi-crystal, bi-epitaxy and so on. In this revision, the existing methods discussed above and those suggested by the referee are properly cited, with short statements summarizing the central concepts.

The revised introduction parts are shown below, while more discussion can be found in the main text: *The precise control of epitaxial lateral homostructures provides additional degrees of freedom to create twisted interfaces/junctions and manipulate topological defects such as phase boundaries⁹, domain walls¹⁰ and vortices¹¹, offering unique platforms for revealing emergent physical properties and developing technically important nanoelectronics. From traditional epitaxial growth point of view, the epitaxial thin films can only be grown following the lattices patterns provided by the chosen single crystal substrates and are constrained to exhibit similar crystal geometries. In this manner, the major advantage of heteroepitaxy using single crystal substrates becomes a shortcoming when it comes to the fabrication of lateral homostructures. To overcome the limitation using single crystal substrates, strategies based on thin film deposition on artificially modified substrates, such as bi(tri)crystalline substrates^{12,13}, step-edge template¹⁴⁻¹⁶, and bi-epitaxy growth¹⁶⁻¹⁹ were proposed. Additionally, hybrid-orientation technology (HOT)²⁰ that utilizes modern semiconductor fabrication procedures including hydrogen implantation, combination of starting wafer and handle wafer, split anneal, and chemical mechanical polishing has also established a landmark for constructing lateral structures composed of different crystal orientations. However, fabricating twisted lateral homostructures with geometry scalability and repetitively alternating configurations remains extremely challenging, especially for strongly correlated systems. In this work, we reveal a generic method assisted by inserting a freestanding layer with controllable twist angle and lattice pattern to fulfill the efficient growth and manipulation of the twisted epitaxial lateral homostructures. The proposed approach paves a platform for the development of lateral homostructures with desired twisted angle, crystal orientation and phase conjunction degrees of freedom, thus opening up a distinct scene for epitaxial growth.*

Reviewer comments:

Another example related to the current manuscript is the hybrid orientation technology (HOT) developed by IBM starting in 2003. Here using wafer bonding between (100)- and (110)-oriented silicon wafers followed by epitaxy, regions of these two desired orientations in which high mobility n-channel and p-channel transistors can be grown are made side-by-side with full lithographic control. A nice overview of this technique is given by Yang, M. *et al.* Hybrid-Orientation Technology (HOT): Opportunities and Challenges. *IEEE Trans. Electron Dev.* **53**, 965–978 (2006).

In conclusion, this manuscript reports nice work, but in my opinion it lacks sufficient novelty to warrant publication in Nature Communications. I also find it concerning and sloppy of the authors to not properly credit those who have done closely related and relevant work prior to them.

Response:

In this response letter, we thank the referee for offering this opportunity to better elucidate the advantages of our proposed approach compared to HOT. The hybrid orientation technology (HOT) developed by IBM is absolutely an elegant process to obtain bicrystal structures. Fabrication of the FinFETs through the HOT requires several procedures including buried oxide (insulator) formation, hydrogen implantation, combination of starting (silicon) wafer and handle wafer, split anneal, polishing, making hybrid orientation array with lithography control, building the FinFETs upon the prepared substrates¹⁴. The conventional HOT processes are schematically illustrated in **Fig. R3**. Among the complex HOT processes, wafer bonding and split annealing are two core factors. Using wafer bonding, the starting wafer and the handle wafer are stacked together through hydrogen bonds, which further becomes strong covalent bonds after heat treatment¹⁵. The next step is to remove the unnecessary part of starting wafer. To do so, hydrogen ions initially implanted into the starting wafer would form hydrogen blisters during high temperature annealing process, and these blisters further leads to the cracking and splitting of the starting wafer^{15,16}. Desired thickness of silicon-on-insulator (SOI) layer can then be obtained through further chemical etching and polishing. After the preparation of substrates, bicrystal arrays composed of Si(001) and Si(110) can then be fabricated through lithography control and CVD.

HOT is a brilliant method to fabricate bi-crystal semiconductor systems, however, it still suffers from some shortcomings and limitations when it comes to the fabrication of complex transition metal oxides. First, in the study done by M. Alexe *et al.* (1999), wafer bonding based on single crystalline oxides, LaAlO₃ and *c*-cut sapphire (Al₂O₃), was demonstrated¹⁶. LaAlO₃ substrates were successfully transferred onto handle wafers, whereas similar method was not applicable for transferring sapphire substrates. Due to relatively high splitting temperature required for wafer splitting, only broken thin strips of sapphire layer are left on the handle wafer. Second, for HOT process, additional chemical mechanical polishing is required to trim the SOI layer down to desired thickness after the splitting of starting wafer. Last, in the process of fabricating the bicrystal array using HOT, it is necessary to use a spacer as an isolation layer to separate the transferred Si from the epitaxial Si, otherwise a defective interface will occur due to the different crystal orientation on both sides¹⁴. These results suggest HOT might not be a generic approach to manufacture complex transition metal oxides.

In the case of our method, no annealing process is required, which eliminates the potential cracks due to split annealing process. The thickness of transferred oxide thin layers can be precisely controlled by using in-situ reflection high energy electron diffraction (RHEED) or other thickness characterizations. This advantage helps to get rid of the polishing process needed in HOT. Last but not least, in our approach, patterned substrates are first prepared, followed by direct epitaxy of desired complex oxide thin films. In this manner, no spacer layer is needed and thus a sharp interface can be obtained, as revealed by transmission electron microscopy shown in revised Fig. 2. These advantages can further reduce the fabrication complexity of the bicrystal structures. Despite the fact that HOT contains many complex fabrication processes, some of which are not suitable for fabricating complex oxides, the HOT demonstrated by IBM has undoubtedly established a landmark for the fabrication of high-efficiency FIN-FETs and twisted semiconductor architectures.

Compared with existing methods including bicrystal, biepitaxial, step-edge and HOT, the main merits of our approach are the substantial simplification of the fabrication process and the generic capability to manufacture many kinds of complex oxide materials with unbounded conjunction tunability. To further strengthen the novelty of our work, in this revision, we further reveal the enhanced second harmonic generation (SHG) using twisted lateral homostructures (please refer to Supplementary Information Fig. S8). Additionally, we demonstrate the alternating orbital control of complex oxides for exploring unconventional physical properties (please refer to revised Fig. 5). These observations have rarely been reported in any existing studies based on twisted lateral homostructures. We sincerely hope that our work could deliver a more efficient and comprehensive strategy to obtain novel twisted lateral homostructures, especially for strongly correlated systems.

Supplementary Information Fig. S8 Second harmonic generation with confocal microscopy mapping of 90° twisted (110)-BFO lateral homostructure:

Revised Fig. 5 Magnetotransport on multijunction LSMO lateral homostructures with alternating orbital configurations:

References

1. Bakaul, S. R. *et al.* Single crystal functional oxides on silicon. *Nat. Commun.* **7**, 10547 (2016).
2. Shen, L. *et al.* Epitaxial lift-off of centimeter-scaled spinel ferrite oxide thin films for flexible electronics. *Adv. Mater.* **29**, 1702411 (2017).
3. Lu, D. *et al.* Synthesis of freestanding single-crystal perovskite films and heterostructures by etching of sacrificial water-soluble layers. *Nat. Mater.* **15**, 1255-1260 (2016).
4. Ji, D. *et al.* Freestanding crystalline oxide perovskites down to the monolayer limit. *Nature* **570**, 87-90 (2019).
5. Chang, Y.-W. *et al.* A fast route towards freestanding single-crystalline oxide thin films by using $\text{YBa}_2\text{Cu}_3\text{O}_{7-x}$ as a sacrificial layer. *Nanoscale Res. Lett.* **15**, 172 (2020).
6. Hilgenkamp, H. & Mannhart, J. Grain boundaries in high- T_c superconductors. *Rev. Mod. Phys.* **74**, 485-549 (2002).
7. Mannhart, J. & Chaudhari, P. High- T_c bicrystal grain boundaries. *Phys. Today* **54**, 48-53 (2001).
8. Marks, R. A., Taylor, S. T., Mammana, E., Gronsky, R. & Glaeser, A. M. Directed assembly of controlled-misorientation bicrystals. *Nat. Mater.* **3**, 682-686 (2004).
9. Char, K., Colclough, M. S., Garrison, S. M., Newman, N. & Zaharchuk, G. Bi-epitaxial grain boundary junctions in $\text{YBa}_2\text{Cu}_3\text{O}_7$. *Appl. Phys. Lett.* **59**, 733-735 (1991).
10. Chiara, A. d. *et al.* $\text{YBa}_2\text{Cu}_3\text{O}_{7-x}$ grain boundary josephson junctions with a MgO seed layer. *IEEE Trans. Appl. Supercond.* **7**, 3327-3330 (1997).
11. Tafuri, F. *et al.* Feasibility of biepitaxial $\text{YBa}_2\text{Cu}_3\text{O}_{7-x}$ josephson junctions for fundamental studies and potential circuit implementation. *Phys. Rev. B* **62**, 14431-14438 (2000).
12. Ramos, J., Ivanov, Z. G., Olsson, E., Zarembinski, S. & Claeson, T. $\text{YBa}_2\text{Cu}_3\text{O}_{7-\delta}$ josephson junctions on directionally ion beam etched MgO substrates. *Appl. Phys. Lett.* **63**, 2141-2143 (1993).
13. Jia, C. L. *et al.* Microstructure of epitaxial $\text{YBa}_2\text{Cu}_3\text{O}_7$ films on step-edge SrTiO_3 substrates. *Phys. C: Supercond.* **175**, 545-554 (1991).
14. Yang, M. *et al.* Hybrid-orientation technology (HOT): Opportunities and challenges. *IEEE Trans. Electron Devices* **53**, 965-978 (2006).
15. Cui, Z. *Wafer bonding* in *Encyclopedia of microfluidics and nanofluidics* (ed Li, D.) 2179-2183 (Springer, New York, 2015).
16. Alexe, M., Kopperschmidt, P., Gösele, U., Tong, Q.-Y. & Huang, L.-J. Wafer bonding involving complex oxides. *Mat. Res. Soc. Symp. Proc.* **574**, 285-292 (1999).

Response to Referee #3

Reviewer comments:

The conventional epitaxial growth of thin films can only control the structure in film thickness direction. In this work, the authors propose a fabrication approach to manipulate the lateral structure by fabricating and twisting free standing substrate film relative to the original substrate, and growing films on the free-standing film and the original substrate. The authors provide detailed characterization results to demonstrate the feasibility of the approach obtain lateral homostructures with different twisted angles and in different types of materials. The work is interesting, but I have several questions regarding their results:

Response:

We sincerely thank the referee for the positive comments on our manuscript. To further the quality of this work, we have modified our manuscript based on the reviewer's comments and suggestions. The detailed changes are addressed in the following sections.

Reviewer comments:

(1) Since in some cases, the misalignment between BFO_{AG} and BFO_{FS} is large, how does the interface accommodate the large alignment? The authors observe the existence of oxygen vacancy in the interfacial region, but oxygen vacancy alone may not accommodate the lattice mismatch between the two parts of the film. Or the two parts of the film just form the physical interface without considering the misalignment. I hope the authors could provide the detailed explanation about the interfacial microstructure.

Response:

We thank the referee for his/her valuable comments and suggestions. We further analyzed the microstructure at the $\text{BFO}_{\text{AG}}\text{-BFO}_{\text{FS}}$ interface shown in Fig. 2, in which we identified that the interface is composed of alternating flat interface and step-like interface regions. A transition monolayer can be clearly seen at the flat interface region. The contrast of this monolayer is weaker than pure Bi columns and stronger than the pure Fe columns in BFO, exhibiting a mixed transition layer. In contrast, a clear atomic step with strong structure distortion can be observed at the step-like interface region. Through the combination of the alternating flat and step-like interfaces at the atomic scale, the large misalignment between BFO_{AG} and BFO_{FS} can be accommodated. We have included more discussion regarding this issue in the revised manuscript, and the Fig. 2 has been revised as follows:

Revised Fig. 2 Microstructure characterization of the 90° twisted BFO lateral homostructure.

a Typical cross-sectional TEM image taken along [001] direction of STO substrate. **b** and **c** HRTEM images of the interface area taken along the [001] and $[1\bar{1}0]$ direction of STO substrate, respectively. **d** HAADF image of the interface area observed along the [001] direction of STO substrate. A gap between FS-STO and STO can be clearly seen, where the space was further filled with BFO. **e** Typical HAADF image of the BFO_{FS}/BFO_{AG} interface. **f** and **g** Enlarged atomic images from the green and red rectangles in (e). **h** O-K and **i** Fe- $L_{2,3}$ edges acquired from the region 1, 2 and 3 in in (e).

The following paragraph is incorporated into the revised manuscript to explain the detailed interface microstructure:

In order to reveal the detailed interfacial microstructure of BFO_{AG} and BFO_{FS} , the interface regions in Fig. 2e were enlarged and overlaid with the structural model of BFO, as shown in Fig. 2f and 2g. A transition monolayer can be seen at the flat interface region, as indicated by the orange dashed box in Fig. 2f. The contrast of this monolayer is weaker than pure Bi columns and is stronger than that of the pure Fe columns in BFO, exhibiting as a mixed transition layer. In contrast, a clear atomic

step is observed in Fig. 2g, as marked by the red circle, where a strong structure distortion can be observed. Through the combination of the alternating flat and step interfaces at the atomic scale, the large misalignment between BFO_{AG} and BFO_{FS} can be accommodated. Figure 2h and 2i show the electron energy-loss near edge structure (ELNES) of O-K and Fe- $L_{2,3}$ from point 1 (BFO_{FS}), point 2 (interface) and point 3 (BFO_{AG}), respectively. The O-K edge from the interface (point 2) clearly shows that the peak a is lower than that of BFO (point 1 and 3), indicating the existence of oxygen vacancy in the interface region. Correspondingly, the Fe- L_3 edge from the interface region show a chemical shift to the lower energy side, as indicated by an arrow in Fig. 2i, indicating that the valence state of Fe decreased at the interface. In addition, atomic resolved X-ray energy dispersive spectroscopy (EDS) mapping of the interface region was performed (see Fig. S4), which further confirmed the high quality BFO homojunction.

Reviewer comments:

(2) The intention of this work is to control the lateral structure of epitaxial film so that some unusual physical properties can be achieved. The authors demonstrate that the polarization and antiferromagnetism of BFO films can be controlled by their proposed fabrication approach, but they didn't show the uniqueness of the approach to obtain unusual multiferroic properties. The nonlinear conductivity is also presented, but the result is also not unique. The demonstration of the effectiveness to achieve unusual physical properties by using this fabrication approach is highly welcomed and this can strengthen the importance of this work.

Response:

We thank the referee for bringing up this issue, offering a key perspective to improve our work further. In response to this, we carried out additional second harmonic generation (SHG) with confocal microscopy mapping and magnetotransport measurements. Through the SHG data shown in the following **Fig. R4** (new Supplementary Information Fig. S8 in the revision), we show that the polarity geometry of the SHG intensity from BFO_{AG} and BFO_{FS} can be arbitrarily controlled by designed twisted angle at the interface, which is expected given that the SHG intensity is sensitive to non-centrosymmetric crystal lattice. However, an unusual increase of nonlinear optical coefficient as well as a significant enhancement of SHG intensity at BFO_{AG}/BFO_{FS} boundary are observed. This can be attributed to the band bending and carrier accumulation due to polarization discontinuity at the twisted interface.

In addition to the SHG measurements, using our fabrication approach, we also demonstrate the effectiveness of achieving periodic orbital alteration, which leads to a direct modulation of magnetotransport behaviors in $(La_{0.7}Sr_{0.3})MnO_3$ homostructures. In this revision, through the fabrication of three devices with different lateral twisted crystalline arrays, we show that the artificially controlled alternating rotation of d -orbitals in $(La_{0.7}Sr_{0.3})MnO_3$ can not only exhibit enhanced magnetoresistance, but also unconventional MR hysteresis (see **Revised Fig. 5**). The results of SHG mapping and magnetotransport highlight the effectiveness to create unconventional physical properties using our fabrication approach. Detailed discussion and corresponding experimental data

of the SHG and magnetotransport are now incorporated into the *supporting information* (Fig. S8) and the *revised main text* (Fig. 5). Once again, we sincerely thank the referee for his/her suggestions to strengthen the importance of our work.

Fig. R4 Second harmonic generation with confocal microscopy mapping of 90° twisted (110)-BFO lateral homostructure. For more details, please refer to Supplementary Information Fig. S8.

Revised Fig. 5 Magnetotransport on multijunction LSMO lateral homostructures with alternating orbital configurations. Please refer to Fig. 5 in the main text for further details.

Reviewer comments:

(3) Other minor issues:

The I-V curve in Fig. 4 could be presented separately to demonstrate the physical outcome of structural manipulation.

The twisted angle should be indicated in the figure caption of the Fig. 2 and Fig. S4.

Response:

We thank the suggestions from the referee. In this revision, the I-V curve and related information in Fig. 4 are presented separately to demonstrate the physical outcome of structural manipulation, as shown the revised Fig. 4 below:

Revised Fig. 4 Artificial creation of BFO polymorph phase boundary via twisted templates. **a** The designer lateral homostructure composed of coexisted tetragonal-like (T-BFO) and rhombohedral-like (R-BFO) BFO phases. Freestanding STO layer (FS-STO) (blue layer) was transferred onto LAO substrate (grey substrate), forming a twisted template providing distinct lattice constrains on both sides. The green and red lattices represent the T-BFO and R-BFO phases, respectively. **b** Surface topography image of the designed lateral homostructure. **c** Cross-section height profile at the dashed yellow line in **(b)** and the enlarged topography images of T-BFO and R-BFO regions. **d-f** Corresponding RSM images around **d** LAO (002), **e** FS-STO (103), and **f** LAO (103) taken from T-R phase boundary, R-BFO, and T-BFO regions, respectively. **g-i** Conductive AFM image **g** taken at the phase boundary with a tip bias of 1.6 V, where the left-bottom inset shows the conductive AFM image taken with a tip bias of -1.6 V. **h** Corresponding cross-section current profile at the dashed yellow line in **(g)**. **i** Corresponding I-V curves taken at T-R boundary (pink), R-BFO (orange), and T-BFO (green), respectively. The scale bar in **c** refer to 500 nm.

Additionally, the information of twisted angle is indicated in the figure caption of the Fig. 2 and Fig. S4:

Fig. 2 Microstructure characterization of the 90° twisted BFO lateral homostructure.

Fig. S4. HAADF-STEM image and EDS mapping of the 90° twisted BFO lateral homostructure.

REVIEWERS' COMMENTS

Reviewer #1 (Remarks to the Author):

The authors have taken all the comments by the referees into consideration and have performed important changes in the manuscript, accordingly. The authors have successfully solved the issues that I mentioned, which were not serious. The other referees raised more important concerns. In particular, referee 2 put forward the missing information about previous reports related to using sacrificial layers as well combined crystal orientations. The authors have acknowledged such omissions and have incorporated new paragraphs on bi-crystal bonding and HOT that are extremely useful to set the current work in perspective. It is clear to me that the ability to tune the conjunction angle at will is a clear advantage of the present method (next to other processing advantages). Still, I agree that the clarification and credit requested by the referee 2 was indeed needed and the manuscript has improved thanks to that. I can therefore advise that it is published in Nat. Comm.

Reviewer #3 (Remarks to the Author):

I don't have further questions. The work can be accepted for publication.